# Effect of Air Particle Abrasion and Primers on Bond Strength to 3D-Printed Crown Materials

**DOI:** 10.3390/ma18020265

**Published:** 2025-01-09

**Authors:** Mohammed Hammamy, Silvia Rojas Rueda, Antonio Pio, Fabio Antonio Piola Rizzante, Nathaniel C. Lawson

**Affiliations:** 1Division of Biomaterials, School of Dentistry, University of Alabama at Birmingham, Birmingham, AL 35209, USA; mhammamy@uab.edu (M.H.); sruedas@uab.edu (S.R.R.); acristal@uab.edu (A.P.); 2Division of Restorative Dentistry, School of Dentistry, Medical University of South Carolina, Charleston, SC 29425, USA; rizzante@musc.edu

**Keywords:** strength, modulus, shrinkage stress, translucency, depth of cure, radiopacity

## Abstract

Two 3D-printed crown materials (Crown and Ceramic Crown) were examined to determine the best surface treatment and primers for bonding. Discs of the two materials were printed and mounted with their “intaglio” surfaces untouched. Half the specimens from each group were sandblasted with 50 µm alumina. Then, specimens were divided into four groups (n = 10): Gr1—no further treatment; Gr2—one coat of silane; Gr3—one coat of universal adhesive; Gr4—one coat of silane, then one coat of universal adhesive. Bond strength specimens were prepared with an Ultradent shear bond strength apparatus using Filtek Supreme composite. Specimens were stored for 8 weeks in 37 °C water. The specimens were debonded with a circular notched-edge blade applied at 1 mm/min, and the shear bond strength was calculated. The data were compared with a two-way ANOVA (factors: surface treatment and primer) and a Tukey post hoc analysis for both materials independently, with *p* < 0.01 considered meaningful. The filler content (burned ash) and resin content (FTIR) of the materials were determined. For both materials, factors surface treatment and primer were significant (*p* < 0.01), but their interaction was not (*p* = 0.43 for Crown and *p* = 0.34 for Ceramic Crown). Alumina air particle abrasion improved the bond strength for both materials. The Tukey post hoc analysis grouped primer treatments into the same statistically different groups for both materials: Gr1 and Gr2 < Gr3 and Gr4. The filler percentage of Crown was 32.7% and Ceramic Crown was 48.2%. Resin content was similar for both materials. The most effective method to bond to 3D-printed crowns (regardless of filler percentage) was to sandblast with 50 µm alumina and apply a layer of adhesive (with or without previous application of silane).

## 1. Introduction

The three-dimensional (3D) printing of dental crowns offers several advantages over milled dental restorations. First, the cost of fabrication of a 3D-printed crown is lower than a milled dental crown due to the lower cost of the both the material and the equipment used to fabricate the crown [1,2]. Second, the 3D-printed crown workflow is more time efficient than a chairside milling workflow, particularly if multiple restorations are fabricated at the same time [1,2]. Finally, the marginal fit and accuracy of 3D-printed restorations is similar to milled ceramic and polymer-based materials [1,2]. There are some limitations of 3D-printed crown materials, such as their strength [3]. The increased flexibility of 3D-printed crowns, however, may offer advantages regarding fatigue performance [4].

Three-dimensionally printed crowns are fabricated from methacrylate-based resins and glass-based filler particles similar to milled resin composite blocks used in milling devices [3]. Milled resin composite blocks suffered a high incidence of clinical debonding (around 10%) leading to one commercially available material losing its indication for full-coverage crowns [5]. The debonding rate was credited to the low modulus of elasticity of the crown materials allowing interfacial stresses at the bond interface with dentin. Finite element analysis testing, however, revealed that the interfacial stresses with dentin for milled resin composite was lower than for lithium disilicate [5]. The interfacial stress with milled resins composite was most problematic when used on titanium or zirconia abutments [5,6]. Therefore, it is possible that the debonding rate of milled resin composite could have been related to the bonding protocol.

The bonding protocol for glass-based ceramics used for chairside restorations involves acid etching and the application of a silane primer [7]. Acid etching milled resin composites removes filler particles and produces a honey-comb texture, whereas alumina particle air abrasion creates a pitted surface [8]. Therefore, most studies recommend alumina air abrasion as an effective step to optimize the bond to milled resin composite [8,9,10,11,12]. Although some studies have reported improved bond strength with the use of a silane primer on milled resin composites [13,14], the use of an adhesive alone or in addition to silane was able to produce a greater bond strength [12,15,16]. Therefore, the differences in the protocol for bonding milled resin composite crowns relative to lithium disilicate crowns may be another reason for the poor clinical bonding performance.

There are several compositional differences between milled resin composites and 3D-printed resin composites that could affect bond strength. First, milled resin composites are polymerized at high temperatures and heat which allows up to a 94.3% degree of conversion [17]. This relatively high degree of conversion has been blamed for reducing the available free monomer for bonding [15]. Filled 3D-printed resin composite also has a high reported degree of conversion (95.2%), whereas unfilled 3D-printed resin composite has a lower degree of conversion (89.1%) [17]. Other studies have reported lower degrees of conversion of 3D-printed crown materials, around 45–65% [18,19]. If a 3D-printed composite has a lower degree of conversion than milled resin composite, then bonding to an available monomer on its surface with an adhesive may improve bond strength. Second, milled resin composites typically contain around 70% ceramic filler, whereas current 3D-printed materials for permanent or temporary crowns contain between 0 and 50% filler [3]. These ceramic filler particles may be composed of silica glass or barium aluminosilicate glass [3]. Bonding to ceramic filler is often accomplished with silane primers. Therefore, bonding to 3D-printed resin composites with a low filler content may not benefit from silane application and may benefit rather from adhesive application. Finally, the monomer composition of 3D-printed crown materials differs from milled resin composites, as 3D-printed composites often contain methacrylate groups not present in milled resin composites, which could have an effect on their bond strength [3].

Previous studies have investigated the effect of surface treatments on the shear bond strength of 3D-printed composite materials used for crowns [20,21,22,23,24,25,26]. Several studies have focused on the effects of alumina particle abrasion and hydrofluoric acid etching on bond strength [20,21,22,23]. Many of these studies concluded that alumina particle abrasion enhanced surface roughness and bond strength more effectively than hydrofluoric acid etching or glass bead abrasion [20,21,22]. Two studies applied air pressure in the range of 2 to 2.5 bar [20,21], while another study found that 4 bar air pressure resulted in better bond strength compared to 1 bar [22]. However, one study that tested crown retention concluded that air particle abrasion did not improve retention strength [23].

Two studies also investigated the use of surface primers and adhesives [21,24]. The first study found that the application of silane—whether with or without prior alumina particle abrasion—improved bond strength to a similar level, which was higher than that of a control group with no surface treatment [21]. The second study reported that applying an adhesive before silane application further enhanced bond strength to 3D-printed composites, although their results were not clearly presented [24].

The objective of this study was to determine the best protocol for bonding to two different 3D-printed crown materials with different filler weight percentages. The materials were characterized for filler weight percentage and resin composition. The intent of characterizing filler content was to determine the proportion of resin available for bonding as well as the confirming the presence of silica-based fillers for bonding. The intent of characterizing the resin content was to confirm the presence of methacrylate groups and determine if there were compositional differences between the two 3D-printed crown materials. As alumina air particle abrasion was reported to improve bonding in previous studies [20,21,22], this surface treatment was examined. Additionally, the use of silane primer was examined due to its success in previous studies [13,14,21]. The novelty of this study was that it also examined the use of an adhesive primer as it has shown favorable bonding with milled resin composites [12,15,16], and 3D-printed crown materials have more resin and a lower degree of conversion than milled resin composites [3,18,19]. The null hypothesis was that there would be no difference in bond strength to either type of 3D-printed crown material with a different surface treatment or primer application.

## 2. Materials and Methods

Two 3D-printed resin composites were chosen for this study (Table 1). A material indicated for temporary crowns was chosen to evaluate the effect of a low-filled material, and a material indicated for permanent crowns was chosen to represent a higher filled material. The characteristics of the materials used in this study as described by their manufacturer are presented in Table 1 as a reference for additional characterization performed in this study.

### 2.1. Filler Characterization

An amount of 5 mL of resin of the two 3D-printed resin composites in Table 1 were placed in an alumina crucible (Coors high-alumina 20 mL crucible, Sigma Aldrich, St. Louis, MO, USA), and their initial weight (W0) was recorded using an analytical balance with precision to 0.0001 g (AE163, Mettler Toledo, Greifensee, Switzerland). To remove the organic matrix, the resin was heated in an electric furnace at 800 °C for 30 min. Afterward, the samples were allowed to cool for 15 min. The remaining inorganic filler was then re-weighed (W1). The filler weight percentage (wt%) was calculated using the formula: Filler wt% = (W1/W0) × 100%.

The materials were then examined using scanning electron microscopy (SEM) to observe the inorganic filler content embedded within the organic resin matrix. Specimens were mounted onto SEM tabs with conductive tape, gold-coated using a vacuum sputter coater and analyzed using secondary electron imaging mode on a SEM (Quanta FEG 650; FEI, Hillsboro, OR, USA). Surface elemental composition of the filler particles was analyzed using energy dispersive spectroscopy (EDS) with an electron energy range of 10–25 keV.

### 2.2. Resin Characterization

The organic resin composition of the two 3D-printed resins listed in Table 1 was analyzed using Fourier transform infrared (FTIR) spectroscopy using an Alpha II ATR-FTIR Spectrometer (diamond crystal, Bruker, Billerica, MA, USA). To enable background subtraction, the bare ATR crystal was first scanned in open air with the press open. Spectra were collected within a 400–4000 cm^−1^ wavelength range, using 16 sample scans and 16 background scans at a resolution of 4 cm^−1^. The spectra were then baseline-corrected in OPUS software (version 8.0) and minimum–maximum normalized. For analysis, the uncured resins were directly applied onto the ATR crystal. Three specimens were prepared for each material, and the spectra were compared to ensure consistency. The FTIR spectra of the resins were also compared to previously reported peaks for pure dental monomers [27].

### 2.3. Shear Bond Strength Testing

Discs (12 mm diameter × 4 mm thickness) were designed in CAD (computer aided design) software (Tinkercad, https://www.tinkercad.com/3d-design, accessed 1 July 2024). The discs were 3D-printed using the two 3D-printed resin composites listed in Table 1 in a vat Digital Light Processing (DLP) printer (Pro 55, SprintRay, Los Angeles, CA, USA). The discs were printed with supports in a 100 µm layer thickness with the bonding surface oriented parallel with the build platform. The discs were cleaned by dipping them into a bowl of 99% ethanol and brushing them until all residue was removed. The discs were thoroughly dried and then cured using their respective program in a cure box (ProCure 2, SprintRay). The disc specimens were then mounted into acrylic blocks with the side of the disc that was not on supports exposed. No polishing was performed on the surface of the disc.

Half the specimens were sandblasted with 50 µm alumina (Cobra, Renfert, Hilzingen, Germany) for 10 s each at 2 bar pressure, wiped with an alcohol wipe and dried. The specimens then received one of four different surface treatments (n = 10/group). In the control group, no further treatment was performed on the specimens. In the silane group, 1 coat of pre-hydrolyzed silane (Porcelain Primer, Bisco, Schaumburg, IL, USA) was applied with a microbrush to the surface of the specimen, allowed to dwell for 30 s, and then air dried. In the adhesive group, 1 coat of universal adhesive (Scotchbond Universal Plus, Solventum, St. Paul, MN, USA) was applied with a microbrush to the surface of the specimen, allowed to react for 20 s, and air dried for 5 s. For the silane and adhesive group, the silane was applied as described previously, and then the adhesive was applied as describe previously.

The specimens were then placed into the Ultradent shear bond strength apparatus (Figure 1). Filtek Supreme composite (Solventum, packable, shade A2) was injected into the white plastic mold inserts. An Elipar S10 curing light (Solventum) with an irradiance of at least 1000 mW/cm^2^ was used and irradiance was confirmed with a radiometer. The curing light was placed in the center of the mold opening and the composite was cured for the complete curing time. After curing, the clamp was lifted off (Figure 1). Specimens were artificially aged by storage in distilled water (without replacement) in an incubator at 37 °C for 3 months.

The specimens were then mounted into a metal vice. The vice was placed into a custom steel fixture which was designed to resist lateral forces during load application. The custom steel fixture contained a circular notched-edge blade for applying load to the button (Figure 1). A load was applied to the custom fixture with a flat indenter in compression using a universal testing machine (Instron 5565, Canton, MA, USA) at a crosshead speed of 1 mm/min (Figure 1). Representative specimens of both materials with and without air particle abrasion were observed using SEM.

Data were checked for normality using Shapiro–Wilk’s test and variance homoscedasticity using Levene’s test using SPSS (IBM, Armonk, NY, USA). Statistical analyses were performed with a level of significance of α = 0.05. Shear bond strength was analyzed separately for each 3D-printed material with a two-way ANOVA for factors surface treatment and primer. An individual 1-way ANOVA and a Tukey post hoc analysis were run when appropriate. A post hoc power analysis was conducted to determine the power of the tests at a significance using G*Power (Heinrich-Heine-Universität, Düsseldorf, Germany).

## 3. Results

The filler weight percentage of Crown was 32.7 +/− 1.4%, and that of Ceramic Crown was 48.2 +/− 2.1%. Representative high magnification SEM images of the filler particles in both materials are shown in Figure 2. The Crown contains irregularly shaped particles composed of barium (Ba), aluminum (Al), silica (Si) and oxygen (O). The Ceramic crown displays larger, irregularly shaped particles along with a sparse distribution of spherical particles. The irregular particles were composed of Si, O, and ytterbium (Yb), and the spherical particles were composed of Si and O, as determined by EDS (Figure 3).

The FTIR spectra of both materials were nearly identical, with a representative FTIR spectrum for each material presented in Figure 4. They both have several peaks to indicate methacrylate groups including peaks at 1636 (C=C), 1319 (C-C), and 1297 (C-C) cm^−1^ [27]. There are slight differences between the resin profiles with Crown possessing peaks 1230 and 1620 cm^−1^ and Ceramic Crown possessing peaks at 1497 and 1600 cm^−1^. Within the fingerprint region of the spectra, there were several peaks similar to bisphenol A-glycidyl methacrylate (Bis-GMA) at 1248, 1297, 1319, 1364, 1380, 1405, 1454, and 1635 cm^−1^, and a strong peak at 1319 cm^−1^ suggesting the presence of triethylene glycol dimethacrylate (TEGDMA) [27].

The mean ± standard deviation of shear bond strength of all groups are presented in Table 2. The post hoc power analysis determined all tests had adequate power (>95%). A two-way ANOVA test determined significant differences for factors surface treatment (*p* < 0.01) and primer (*p* < 0.01) for both materials. The interaction of surface treatment*primer was not significant for Crown (*p* = 0.43) or Ceramic Crown (*p* = 0.34). Alumina air particle abrasion improved the bond strength for both materials. The Tukey post hoc analysis grouped primer treatments into the same statistically different groups for both materials, as indicated in Table 2.

Representative SEM images of Crown and Ceramic Crown following air particle abrasion demonstrate their surface textures (Figure 5 and Figure 6).

## 4. Discussion

The objective of this study was to determine an optimal bonding protocol for both a temporary and permanent 3D-printed crown material. The results of the study indicated that the highest bond strength for both materials could be achieved by air particle abrasion followed by adhesive application alone or silane and adhesive application. Therefore, the null hypothesis was rejected.

The results of this study confirm the findings of previous studies which have reported a higher shear bond strength to a permanent 3D-printed crown material following alumina air particle abrasion [20,21,22]. The air pressure used in the current study (2 bar) is similar those reported in previous studies (2–4 bar). At these pressures, alumina air particle abrasion was found to increase surface roughness (arithmetic average roughness, Ra), as measured by a profilometer from 0.2 to 0.5 µm for the untreated group to 2.3–3.8 µm for the particle abraded group [20,21]. Unlike previous studies, the groups which did not receive surface treatment in this study were only wiped with an alcohol wipe following the washing and curing procedures. In previous studies, the surfaces of the “untreated” samples were polished with silicon carbide to produce a standardized surface [20,21]. This condition does not represent the clinical scenario in which the intaglio surface of the restoration would remain in an as-printed condition without further surface treatment.

Another factor which can influence the initial surface texture of a 3D-printed material is the orientation at which it is printed relative to the build plate. In this study, specimens were printed parallel (0°) to the build platform. A previous study determined that printing at a 45° to the build platform produced a greater roughness than printing at 0° or 90° to the build platform; however, this roughness did not significantly affect shear bond strength to the 3D-printed crown material [25].

A finding observed with some specimens was a white residue present on the bonding surface of the specimens that could be observed following the washing procedure (Figure 7). If this residue was observed, the specimens were discarded. An observation of this residue under SEM reveals that the residue is composed of excess filler particles accumulated on the surface of the specimen (Figure 8). This accumulation is speculated to occur if prolonged washing with ethanol removes an excessive amount of resin and exposes filler. A previous study has reported a significant decrease in bond strength when 3D-printed permanent and temporary crown materials were washed in ethanol for 10 min, 1 h or 8 h relative to the suggested 5 min cleaning time [26]. The decrease in bond strength was credited to the elution of resin and filler leading to a decrease in surface roughness or weakening of the surface of the 3D-printed material and cohesive bond failures.

The universal adhesive proved more effective than the silane primer for bonding to both types of 3D-printed crown materials. These findings align with previous studies that have demonstrated enhanced shear bond strength to milled resin composites when using adhesive primers [12,15,16]. A study of bonding to a 3D-printed crown material used visio.link primer, which contains methyl methacrylate, dimethacrylate, and pentaerythritol acrylate [22]. The present study utilized a universal adhesive incorporating methacrylate resins, MDP, and silane. The primary aim of this study was to assess whether the methacrylate in the adhesive could improve bond strength to 3D-printed resin composites, though other components of the adhesive may have also contributed to its effectiveness. MDP, a molecule known for bonding to zirconia, and silane, which bonds to silica glass, were both present in the adhesive. However, since the 3D-printed crown materials did not contain zirconia fillers, it is unlikely that MDP played a significant role in the adhesion. While the silane in the adhesive could have facilitated chemical bonding to the silica-based filler particles in the 3D-printed materials, its contribution is likely minimal, as the use of a pure silane primer did not result in a significant improvement in adhesion.

The favorable bond achieved when using the adhesive primer may be due to the presence of a large amount of resin in the 3D-printed materials. The relatively lower degree of conversion reported for 3D-printed materials may allow more available monomer for bonding [18,19]. Although the paper did not aim to compare the bond strength to the temporary crown material and permanent crown material, the bond strength values appear higher for the temporary crown material. The temporary crown material contained approximately 67% resin, whereas the permanent crown contained only 52% resin, so there may have been more available methacrylate groups available for bonding in the temporary crown material. It is also possible that differences in bond strength are related to the slight differences in monomer composition between the temporary and permanent crown materials as observed in FTIR.

There were several limitations of this study, including the lack of standardization of the bonding surface with polishing, lack of thermal or mechanical fatigue, and limited number of materials examined. Although the intent of the study was to allow bonding to an as-printed intaglio surface, the standardization of the bonding surface without silicon carbide polishing is difficult. This study simulated aging by long-term water storage rather than thermocycling, which would have provided more stress on the bond interface. The temporary crown material used in this study is no longer commercially available; however, the information gleaned from this material may be similar to other low-filled 3D-printed temporary crown materials. New resins for 3D-printing are constantly updated, and therefore, independent research of resins may lag behind the commercial release of new products.

Future studies should confirm the degree of conversion of 3D-printed crown materials and determine if conditions which vary the degree of conversion (i.e., extra curing cycles, curing in glycerin, use of different curing units, etc.) influence bond strength. Different 3D-printed resin composites with other filler and resin composition, as well as different primers, should be evaluated.

## 5. Conclusions

In conclusion, this study successfully identified an optimal bonding protocol for both temporary and permanent 3D-printed crown materials, demonstrating that air particle abrasion followed by adhesive application, either alone or in combination with silane, resulted in the highest bond strength. These findings are consistent with previous research, particularly regarding the use of alumina air particle abrasion to enhance bond strength. The high shear bond strength achieved with these materials with the use of an adhesive is speculated to occur due to the high proportion of methacrylate-based resin in the 3D-printed resin composites. Although bonding to the silica-based ceramic fillers likely occurred with the silane primer, it was not as effective as the bond enhancement achieved with the adhesive primer. The study also highlighted the potential impact of material composition, such as the higher resin content in the temporary crown material, which may contribute to stronger bond strengths.

## Figures and Tables

**Figure 1 materials-18-00265-f001:**
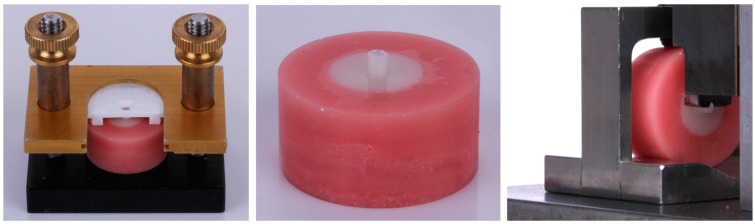
Ultradent shear bond strength apparatus (**left**); specimen with composite resin button on the surface of the 3D-printed disc (**center**); circular notched-edge blade for applying load to the button (**right**).

**Figure 2 materials-18-00265-f002:**
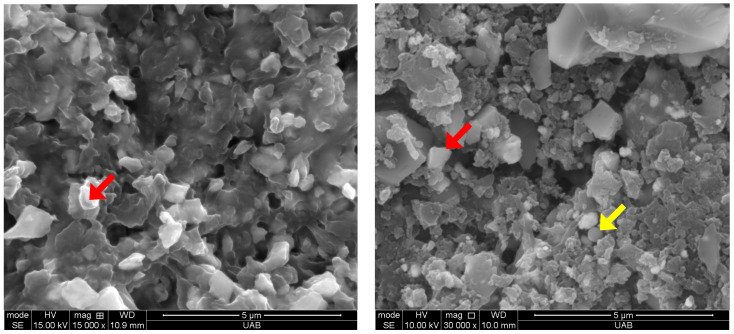
Filler particles of the materials used in this study: (**left**) Crown, (**right**) Ceramic Crown. Examples of an irregular-shaped particles indicated with red arrows and spherical particle indicated with yellow arrow. SEM was taken prior to surface treatment.

**Figure 3 materials-18-00265-f003:**
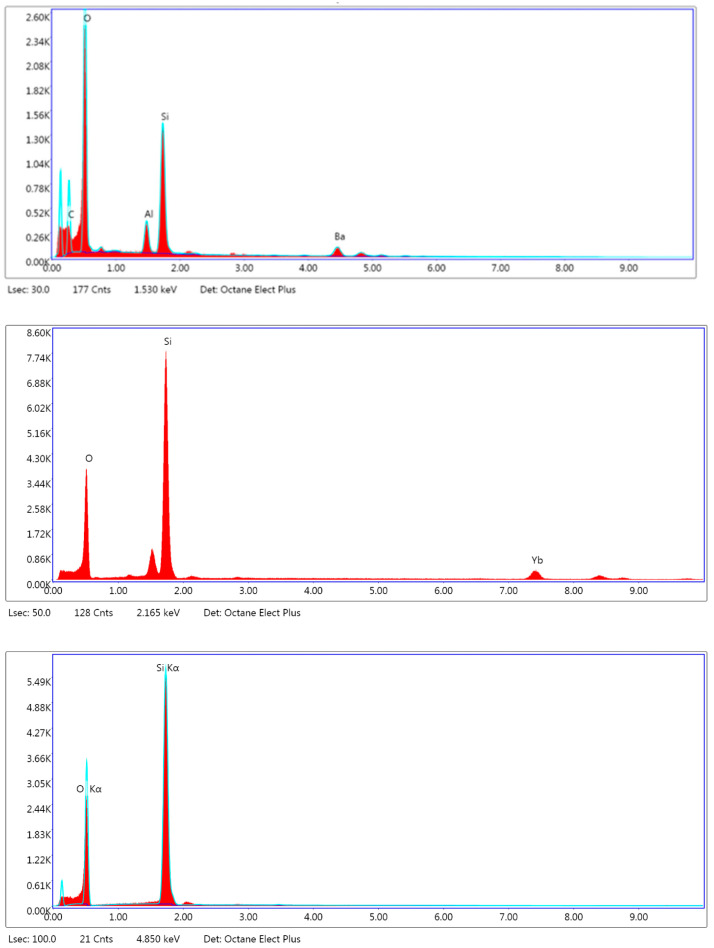
EDS spectra of (**top**) irregular filler particles in Crown, (**middle**) irregular filler particles Ceramic Crown, and (**bottom**) spherical particles in Ceramic Crown.

**Figure 4 materials-18-00265-f004:**
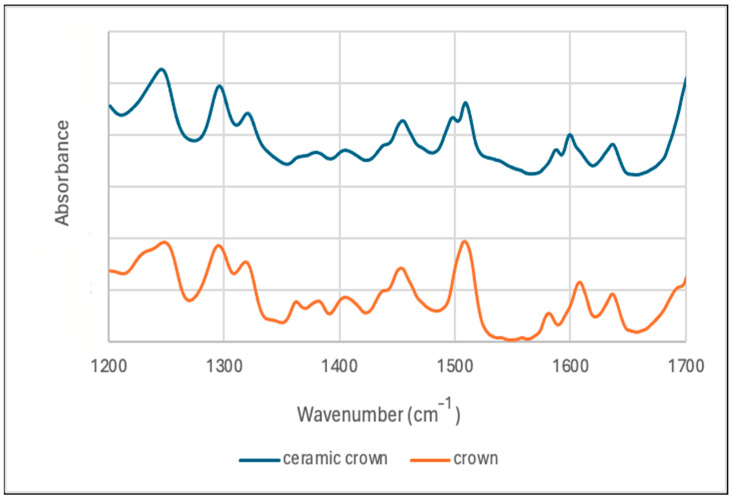
FTIR spectrum of the materials used in this study.

**Figure 5 materials-18-00265-f005:**
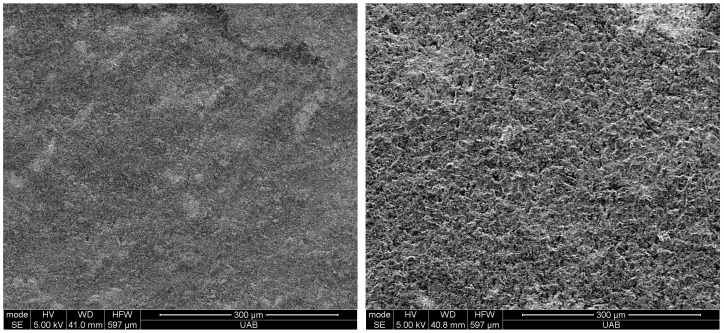
Surface of Crown material (**left**) without particle abrasion, and (**right**) with particle abrasion.

**Figure 6 materials-18-00265-f006:**
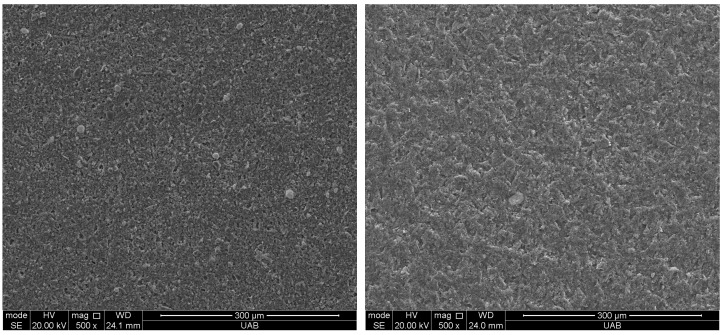
Surface of Ceramic Crown material (**left**) without particle abrasion, and (**right**) with particle abrasion.

**Figure 7 materials-18-00265-f007:**
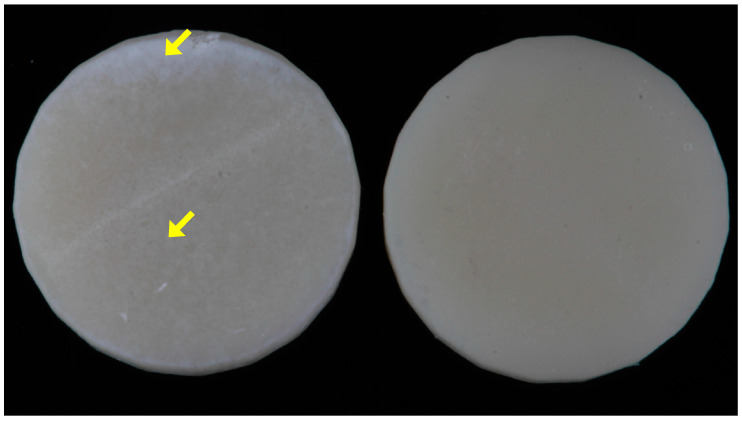
White residue (yellow arrow) present on some discarded specimens of Ceramic Crown following ethanol cleaning (**left**) and Ceramic Crown specimen without white residue (**right**).

**Figure 8 materials-18-00265-f008:**
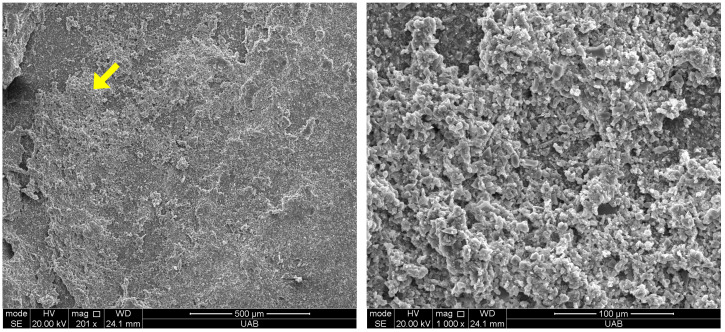
Microscopic view (200×) of white residue (yellow arrow) which appears to be loose filler particles (**left**), and higher magnification view (1000×) of the loose fillers (**right**).

**Table 1 materials-18-00265-t001:** Materials used in this study.

Material	Manufacturer	Classification	Composition *
Crown	SprintRay, Los Angeles, CA, USA	Temporary 3D-printed crown resin composite material	Proprietary oligomers, monomers, photoinitiators, additives
Ceramic Crown	SprintRay	Permanent 3D-printed crown resin composite material	Proprietary oligomers, monomers, photoinitiators, additives
Porcelain Primer	Bisco, Schaumburg, IL, USA	Silane primer	Ethanol, acetone, silane
Scotchbond Universal Plus	Solventum, St Paul, MN, USA	Universal adhesive primer	Dimethacrylate resins, MDP phosphate monomer, silane mixture, HEMA, Vitrebond copolymer, filler, ethanol, water, initiators, dual-cure accelerator
Cobra	Renfert, Hilzingen, Germany	Alumina particles	50 µm aluminum oxide

MDP = Methacryloyloxydecyl Dihydrogen Phosphate; HEMA = Hydroxyethyl Methacrylate; * Reported by manufacturer.

**Table 2 materials-18-00265-t002:** Mean ± standard deviation of shear bond strength (MPa). Groups in each column with different letters are statistically significantly different as determined by Tukey post hoc analysis. Groups with alumina air particle abrasion were significantly different than those with no surface treatment for each type of primer as determined by 2-way ANOVA (*p* < 0.01).

	Crown		Ceramic Crown	
	No Surface Treatment	Alumina Air Particle Abrasion	No Surface Treatment	Alumina Air Particle Abrasion
No primer	16.3 ± 3.9 ^a^	24.4 ± 7.5 ^a^	6.9 ± 5.5 ^a^	17.3 ± 3.6 ^a^
Silane	16.2 ± 3.7 ^a^	24.1 ± 4.7 ^a^	6.8 ± 4.5 ^a^	18.7 ± 5.6 ^a^
Adhesive	34.4 ± 5.5 ^b^	48 ± 5.7 ^b^	19.5 ± 9.8 ^b^	35.4 ± 3 ^b^
Silane + Adhesive	39.2 ± 5.9 ^b^	47.8 ± 6.2 ^b^	25.6 ± 7.1 ^b^	33.3 ± 7.1 ^b^

## Data Availability

The original contributions presented in this study are included in the article. Further inquiries can be directed to the corresponding author.

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
