# Peer review of "Effect of Air Particle Abrasion and Primers on Bond Strength to 3D-Printed Crown Materials"

_materials, 2025, doi:10.3390/ma18020265_

Round 1
Reviewer 1 Report
Comments and Suggestions for Authors
In the manuscript, the authors studied the effects of different surface treatment and primers on bond strength to 3D-printed crown materials with different compositions. Two printable crown materials and four combinations of surface treatment and primers were used in this study. Two statistical analysis approaches were applied to analyze the data. Based on the result, the authors draw a conclusion that sandblasting with 50 um alumina and applying the adhesive coating was the most effective method to bond a 3D-printed crown. Overall, the manuscript presented an interesting study and might benefit the applications of 3D printing in dental restoration. Therefore, the reviewer recommends this manuscript to be published after addressing some questions. Here are the suggestions and questions for the authors:
1. In the first sentence of abstract, it should be “compositions” instead of “composition”.
2. In Table 1, please also explain the full form of HEMA.
3. In the first paragraph of Results section, the authors mentioned spherical particles. However, the reviewer was not sure which particles can be identified as spherical particles since all the particles looked irregularly shaped. The reviewer suggested the authors mark them in the Figure.
4. The authors claimed different elements in the materials shown in the SEM images. The reviewer recommended the authors to add EDX images to make the claim more convincing.
5. How many replicates did the authors have for each group of surface treatment and coating?
6. In Figure 5, which part is the “white residue”? Please mark it in the figure.
Author Response
- In the first sentence of abstract, it should be “compositions” instead of “composition”.
Response : Thank you for pointing this out. Based on the comments of another reviewer, the Abstract was shortened and this sentence was removed.
- In Table 1, please also explain the full form of HEMA.
Response: Thank you for pointing this out. We have added this information below Table 1.
- In the first paragraph of Results section, the authors mentioned spherical particles. However, the reviewer was not sure which particles can be identified as spherical particles since all the particles looked irregularly shaped. The reviewer suggested the authors mark them in the Figure.
Response: Agree. We have added colored arrows in Figure 2.
- The authors claimed different elements in the materials shown in the SEM images. The reviewer recommended the authors to add EDX images to make the claim more convincing.
Response: We agree. We have added the EDS Spectra as Figure 3.
- How many replicates did the authors have for each group of surface treatment and coating?
Response: Thank you for pointing out this missing information. The sample size was 10/group. We have added this information to the Materials and Methods -> Shear Bond Strength Testing -> Paragraph 2 -> Sentence 2.
- In Figure 5, which part is the “white residue”? Please mark it in the figure.
Response: We understand the confusion. We have included a photographic picture of the residue to better explain the nature in Figure 6.
Reviewer 2 Report
Comments and Suggestions for Authors
The article titled "Effect of Air Particle Abrasion and Primers on Bond Strength to 3D-Printed Crown Materials" is well-written and highly engaging. The topic is relevant and contributes valuable insights to the field. The English language is clear and professional, making the content easy to understand. Apart from a few minor suggestions for improvement, I believe the article is ready for publication.
In the Table 1, the manufacturer should be listed formally, including the full company name, city, and country. For example: SprintRay Inc., Los Angeles, CA, USA.
Please include the P-values and specify the statistical test used in Table 2 for clarity and better interpretation of the results.
Also, in the Table 1 instead of using the "+/- " symbol, use "±" symbol for standard deviation, please present the standard deviation values in parentheses.
Author Response
In the Table 1, the manufacturer should be listed formally, including the full company name, city, and country. For example: SprintRay Inc., Los Angeles, CA, USA.
Response: Thank you for pointing this out. We have added this information below Table 1.
Please include the P-values and specify the statistical test used in Table 2 for clarity and better interpretation of the results.
Response: Thank you for this recommendation. The following information has been added to the Figure 2 legend: “…as determined by Tukey post-poc analysis. Groups with alumina air particle abrasion were significantly different than those with no surface treatment for each type of primer as determined by 2-way ANOVA (p < 0.01).”
Also, in the Table 1 instead of using the "+/- " symbol, use "±" symbol for standard deviation, please present the standard deviation values in parentheses.
Response: We agree. We have changed the "+/- " symbol to the "±" symbol. We understand that some journals prefer to represent data as “average ±standard deviation” and others as “average (standard deviation)”, however, I have never seen “average ±(standard deviation)”. However, if this is required, we can accommodate.
Reviewer 3 Report
Comments and Suggestions for Authors
The abstract is too long. There is too much information concerning the formation of materials and the storage conditions. Please check the regulations for the Authors. Indicate which of the groups of the obtained materials seems to be the most promising.
“Introduction”: “The objective of this study was to evaluate the bond strength to two different 3Dprinted crown materials with different filler weight percentages “. The mentioned sentence does not correspond with the title of the manuscript.
In the ”Introduction” section, there is no information concerning different types of fillers. Moreover, there is no information on what type of filler was used in the submitted manuscript.
“Introduction”: Grammar tense is not consistent. Both past and future tenses appear in this section.
Statements formulated in the following passage need relevant references:“ As alumina air particle abrasion has been reported to improve bonding in previous studies, this surface treatment will be examined. Additionally, the use of silane primer will be examined due to its success in previous studies. “
Table 1: Presenting the composition of the used materials is insufficient. It is not possible to explain the reactions and reasons why particular bonds have formed.
The reference to Table 1 in the sentence “5mL of resin of the two 3D-printed resin composites in Table 1 ..” has apparently been made by mistake since Table 1 does not present the composition of resin composites!
Please describe the composition as well as formation process of the materials mentioned in the sentence: “Two 3D-printed resin composites were chosen for this study. “
In the case of the FTIR method, a spectrometer is used, not a spectrophotometer! Moreover, information concerning the type of crystal used during FTIR-ATR measurements needs to be included in the text.
Sentence: “Filler weight percentage of Crown was 32.7 +/-1.4%, and Ceramic Crown was 48.2 +/- 2.1%.“ What type of filler was used in the studied materials? Please use adequate methods to analyze the fillers.
Sentence: “Crown contains irregularly shaped particles composed of barium (Ba), aluminum (Al), silica (Si) and oxygen (O). “ (page 5). Explain how the observations mentioned above were made.
Sentence: “Representative SEM images of the materials are shown in Figure 2.“ Please explain at which stage of the analysis the images were recorded.
Sentence: “Filler weight percentage of Crown was 32.7 +/-1.4% and Ceramic Crown was 48.2 +/- 2.1%“. The SEM images of the filler should be included in the work.
FTIR analysis: Please correlate all of the mentioned bands with appropriate groups and vibrations present in the studied materials. Moreover, Figure 3 has to be improved. The Y-axis should not have numeral designations.
In the case of the FTIR bands, no peaks are analyzed. Please provide relevant references in the text.
Table 2: The numbers of particular groups have to be added. Moreover, the columns with changes (results of subtraction: Alumina air particle abrasion - No surface treatment)
Please add SEM images of all materials before and after measurements.
How was the surface roughness calculated? Explain the meaning of Ra.
Sentence: “A finding observed with some specimens was a white residue present on the bonding surface of the specimens.“ Please provide images of the “white residue“; magnification of the analysed surfaces has to be added.
Sentence: “These findings align with previous studies that have demonstrated enhanced bond strength.“ The reference is ambiguous since no specific bonds have been indicated.
Sentence: “There were several limitations of this study.“ Please indicate those limitations in the text.
Sentence: “The temporary crown material used in this study is no longer commercially available.“ What was the reason to analyze a material that is not available?
Conclusions. The Authors present observations rather than conclusions. Explain the reason why the adhesive and silane improve bond strength.
The manuscript has to be rearranged, and the SEM images need to be included, as well as the description of the FTR results, compositions of the used components, and an explanation of changes in bond strength.
Author Response
- The abstract is too long. There is too much information concerning the formation of materials and the storage conditions. Please check the regulations for the Authors.
Response: Thank you for pointing this out. We have shortened it.
- Indicate which of the groups of the obtained materials seems to be the most promising.
Response: The objective of this study has been more clearly defined. We did not intend to compare the bond between a temporary crown material and permanent crown material. We wanted to find the optimal protocol for bonding to these materials. The most promising protocol is listed in the last sentence of the Abstract : “The most effective method to bond to 3D-printed crowns (regardless of filler percentage) was to sandblast with 50 µm alumina and apply a layer of adhesive (with or without previous application of silane).”
- “Introduction”: “The objective of this study was to evaluate the bond strength to two different 3Dprinted crown materials with different filler weight percentages “. The mentioned sentence does not correspond with the title of the manuscript.
Response: You are correct. We have changed this sentence to “The objective of this study was to determine the best protocol for bonding to two different 3D-printed crown materials with different filler weight percentages.”
- In the ”Introduction” section, there is no information concerning different types of fillers. Moreover, there is no information on what type of filler was used in the submitted manuscript.
Response: Thank you for this recommendation. This sentence has been added to the 4th paragraph of the Introduction: “These ceramic filler particles may be composed of silica glass or barium aluminosilicate glass [3].” The filler type was not listed in Table 1 as this is proprietary information not reported by the manufacturer. We have attempted to characterize the fillers by SEM and EDS analysis as presented in the paper. We have attempted to more clearly explain this analysis in the Results section.
- “Introduction”: Grammar tense is not consistent. Both past and future tenses appear in this section.
Response: Great point. We have corrected this section so that it uses only past tense.
- Statements formulated in the following passage need relevant references:“ As alumina air particle abrasion has been reported to improve bonding in previous studies, this surface treatment will be examined. Additionally, the use of silane primer will be examined due to its success in previous studies. “
Response: Thank you. We have added the following references: “As alumina air particle abrasion was reported to improve bonding in previous studies [20-22], this surface treatment was examined. Additionally, the use of silane primer was examined due to its success in previous studies [13,14,21]. The novelty of this study was that it also examined the use of an adhesive primer as it has shown favorable bonding with milled resin composites [12,15,16] and 3D-printed crown materials have more resin and a lower degree of conversion than milled resin composites [3,18,19].”
- Table 1: Presenting the composition of the used materials is insufficient. It is not possible to explain the reactions and reasons why particular bonds have formed.
Response: Thank you. The compositions of the materials presented in Table 1 represent the information provided by the manufacturer. We agree that information is insufficient to give a rationale for certain bonds. Therefore, we conducted additional characterization of the resin and filler components of the materials. This sentence was added at the end of the first paragraph of the Materials and Methods section: “The characteristics of the materials used in this study as described by their manufacturer are presented in Table 1 as a reference for additional characterization performed in this study.”
- The reference to Table 1 in the sentence “5mL of resin of the two 3D-printed resin composites in Table 1 ..” has apparently been made by mistake since Table 1 does not present the composition of resin composites!
Response: Thank you and I understand the confusion. The 3D-printed materials used in this study are resin with filler (also called “resin composite”). I have modified the third column of Table 1 to indicate that the crown materials are resin composites.
- Please describe the composition as well as formation process of the materials mentioned in the sentence: “Two 3D-printed resin composites were chosen for this study. “
Response: Thank you for pointing that out. The characteristics of these materials are presented in Table 1 which describes their formation process (3D-printing) and composition as described by the manufacturer. We have added “(Table 1)” at the end of the sentence quoted by the reviewer to make that more clear. Additionally, we have further characterized these materials through FTIR, SEM and EDS.
- In the case of the FTIR method, a spectrometer is used, not a spectrophotometer! Moreover, information concerning the type of crystal used during FTIR-ATR measurements needs to be included in the text.
Response: Thank you for pointing that out. Spectrophotometer was changed to spectrometer. The type of crystal (diamond) was added to the first sentence of the “Resin Characterization” section of the Materials and Methods.
- Sentence: “Filler weight percentage of Crown was 32.7 +/-1.4%, and Ceramic Crown was 48.2 +/- 2.1%.“ What type of filler was used in the studied materials? Please use adequate methods to analyze the fillers.
Response: Thank you for this comment. The filler particles have been observed by SEM and elemental composition determined by EDS. “…as determined by EDS (Figure 3).” was added to the last sentence of the first paragraph of the Results section. Additionally, Figure 3 was added to show the results of EDS testing.
- Sentence: “Crown contains irregularly shaped particles composed of barium (Ba), aluminum (Al), silica (Si) and oxygen (O). “ (page 5). Explain how the observations mentioned above were made.
Response: Thank you for this comment. The elemental composition was determined by EDS. “…as determined by EDS (Figure 3).” was added to the last sentence of the first paragraph of the Results section. Additionally, Figure 3 was added to show the results of EDS testing.
- Sentence: “Representative SEM images of the materials are shown in Figure 2.“ Please explain at which stage of the analysis the images were recorded.
Response: Thank you for pointing this out. Figure 2 is a high magnification (indicated by scale bar) image of the materials used to demonstrate the shape of the filler particles. We have added “SEM was taken prior to surface treatment.” to the end of Figure 2 legend. We have added “Representative high magnification SEM images” to the second sentence of the Results section.
- Sentence: “Filler weight percentage of Crown was 32.7 +/-1.4% and Ceramic Crown was 48.2 +/- 2.1%“. The SEM images of the filler should be included in the work.
Response: Thank you for pointing this out. Figure 2 demonstrates the shape of the filler particles. The Figure has been relabeled to “Figure 2. Filler particles of the materials used in this study (left) Crown, (right) Ceramic Crown. Examples of an irregular-shaped particles indicated with red arrows and spherical particle indicated with yellow arrow”
- FTIR analysis: Please correlate all of the mentioned bands with appropriate groups and vibrations present in the studied materials. Moreover, Figure 3 has to be improved. The Y-axis should not have numeral designations.
Response: Thank you for this comment. We have modified the Introduction (last paragraph, 3rd sentence) to include this information: “The intent of characterizing filler content was to determine the proportion of resin available for bonding as well as the confirming the presence of silica-based fillers for bonding. The intent of characterizing the resin content was to confirm the presence of methacrylate groups and determine if there were compositional differences between the two 3D-printed crown materials.” This information was added for the reader to understand that we did not want to identify the composition of the resin, only confirm that there were methacrylate groups present for bonding. We have identified the peaks that correspond to methacrylate groups in the Results section (second paragraph, second sentence): “They both have several peaks to indicate methacrylate groups including peaks at 1636 (C=C), 1319 (C-C), and 1297 (C-C) cm⁻¹ [27].” Additionally, we have included which peaks correspond to known molecules within the fingerprint region of the spectrum (Results, 2 paragraph, 4th sentence): “Within the fingerprint region of the spectra, there were several peaks similar to bisphenol A-glycidyl methacrylate (Bis-GMA) at 1248, 1297, 1319, 1364, 1380, 1405, 1454, and 1635 cm⁻¹, and a strong peak at 1319 cm⁻¹ suggesting the presence of triethylene glycol di-methacrylate (TEGDMA) [27].”. The y-axis labels have also been removed.
- In the case of the FTIR bands, no peaks are analyzed. Please provide relevant references in the text.
Response: Thank you for this comment. We have identified the relevant peaks to suggest the presence of methacrylate groups in the resin. Additionally, we have provided #29 as our FTIR spectrum reference for known molecules in the fingerprint region.
- Table 2: The numbers of particular groups have to be added. Moreover, the columns with changes (results of subtraction: Alumina air particle abrasion - No surface treatment)
Response: Thank you for pointing this out. We have used n=10 for the entire study. We have added this information to the Materials and Methods -> Shear Bond Strength Testing -> Paragraph 2 -> Sentence 2.
- Please add SEM images of all materials before and after measurements.
Response: Thank you for this recommendation. Additional SEMs have been added of the Ceramic Crown group (Figure 6).
- How was the surface roughness calculated? Explain the meaning of Ra.
Response: Thank you for mentioning this. Ra stands for arithmetic average roughness. Roughness is measured using a profilometer. This information was added to the third sentence of the second paragraph of the Discussion section.
- Sentence: “A finding observed with some specimens was a white residue present on the bonding surface of the specimens.“ Please provide images of the “white residue“; magnification of the analysed surfaces has to be added.
Response: Thank you for this recommendation. A photograph of the residue was included. Additionally, better SEMs were included that labeled the residue. And the magnifications of the SEMs were included. - Sentence: “These findings align with previous studies that have demonstrated enhanced bond strength.“ The reference is ambiguous since no specific bonds have been indicated.
Response: Thank you for this point. This sentence has been modified to “These findings align with previous studies that have demonstrated enhanced shear bond strength” to reinforce the fact that this study measured shear bonding between two surfaces as an applied engineering endeavor rather than chemical bonding.
- Sentence: “There were several limitations of this study.“ Please indicate those limitations in the text.
Response: Thank you for this point. The exact limitations (lack of standardization of bonding surface with polishing, lack of thermal or mechanical fatigue, and limited number of materials examined) were explicitly listed at the end of this sentence to make the point more obvious.
- Sentence: “The temporary crown material used in this study is no longer commercially available.“ What was the reason to analyze a material that is not available?
Response: Thank you for this point. The following information was added at the end of that sentence “…, however, the information gleaned from this material may be similar to other low-filled 3D-printed temporary crown materials.”
- Conclusions. The Authors present observations rather than conclusions. Explain the reason why the adhesive and silane improve bond strength.
Response: This is an excellent point. The following sentences have been added to the Conclusion: The high shear bond strength achieved with these materials with the use of an adhesive is speculated to occur due to the high proportion of methacrylate-based resin in the 3D-printed resin composites. Although bonding to the silica-based ceramic fillers likely occurred with the silane primer, it was not as effective as the bond enhancement achieve with the adhesive primer.
Round 2
Reviewer 3 Report
Comments and Suggestions for Authors
I am highly impressed that the Authors managed to address and supplement the text of the publication with all the additional information and results. Moreover, the discussion has been expanded, which significantly enhances the value of the manuscript.